# A Green Method of Extracting and Recovering Flavonoids from *Acanthopanax senticosus* Using Deep Eutectic Solvents

**DOI:** 10.3390/molecules27030923

**Published:** 2022-01-29

**Authors:** Xinyu Zhang, Jianqing Su, Xiuling Chu, Xiaoya Wang

**Affiliations:** College of Agronomy, Liaocheng University, Liaocheng 252000, China; xinxinxinxinyuyuyu@163.com (X.Z.); wangxy9625@163.com (X.W.)

**Keywords:** deep eutectic solvent, *Acanthopanax senticosus*, flavonoids, extraction, macroporous resins, recycle

## Abstract

In recent years, green extraction of bioactive compounds from herbal medicines has generated widespread interest. Deep eutectic solvents (DES) have widely replaced traditional organic solvents in the extraction process. In this study, the efficiencies of eight DESs in extracting flavonoids from *Acanthopanax senticosus* (AS) were compared. Response surface methodology (RSM) was employed to optimize the independent variable including ultrasonic power, water content, solid-liquid ratio, extraction temperature, and extraction time. DES composed of glycerol and levulinic acid (1:1) was chosen as the most suitable extraction medium. Optimal conditions were ultrasonic power of 500 W, water content of 28%, solid-liquid ratio of 1:18 g·mL^−1^, extraction temperature of 55 °C, and extraction time of 73 min. The extraction yield of total flavonoids reached 23.928 ± 0.071 mg·g^−1^, which was 40.7% higher compared with ultrasonic-assisted ethanol extraction. Macroporous resin (D-101, HPD-600, S-8 and AB-8) was used to recover flavonoids from extracts. The AB-8 resin showed higher adsorption/desorption performance, with a recovery rate of total flavonoids of up to 71.56 ± 0.256%. In addition, DES solvent could efficiently be reused twice. In summary, ultrasonic-assisted DES combined with the macroporous resin enrichment method is exceptionally effective in recovering flavonoids from AS, and provides a promising environmentally friendly and recyclable strategy for flavonoid extraction from natural plant sources.

## 1. Introduction

*Acanthopanax senticosus*, also known as Siberian ginseng in China, is widely distributed in Russia and Asia [1]. The plant has been traditionally used for nourishing the liver and kidney, strengthening muscles and bones, improving kidney function, and soothing the nerves [2]. All parts of the plant, including its roots, stems, leaves, flowers, and fruits, have pharmacological activities, and are often used as functional medicine [3]. Active ingredients of the plant components, such as flavonoids, have been shown to improve the body’s innate immunity and antioxidant capacity [4,5], reduce myocardial cell damage [6], and enhance antibacterial efficacy [7], among other effects. AS flavonoids identified to date include hyperoside, quercetin, citrin, and rutin [8]. Conventional solvent extraction, using chemicals such as methanol, ethanol [9], and ethyl acetate, was previously used. However, organic solvents are expensive, flammable, toxic, and non-degradable, and their excessive use not only damages the natural environment, but also threatens human health [10]. Several non-toxic or less toxic alternatives have been explored, such as heterogeneous catalysts, supercritical fluids, and ionic liquids (ILS) [11]. Among them, ILs are composed of multiple components with characteristic ionic bonds formed between cations and anions. The extraction of flavonoids by this method has attracted attention, and it has the advantages of clean process and high extraction efficiency [12]. However, one notable limitation is that many ILs are based on imidazole and pyridine, and therefore exert toxic effects [13]. Therefore, a “green solvent” that is easy to operate, efficient, and consumes less energy for extraction of AS total flavonoids remains a critical requirement.

Eutectic solvents were initially synthesized in 2001 [14], and have been shown to possess similar physical and chemical properties as ILs. In 2011, the concept of “natural eutectic solvents” (NADES) was proposed. These solvents usually contain secondary metabolites, such as sugar alcohols, sugars, amino acids, and organic acids, as NADES components. Traditional ionic liquids are mostly prepared through ion exchange or neutralization reactions, which involve issues such as complex preparation, high cost, and difficulty in purification, and the toxicity of pyridine and imidazole ionic liquids is even comparable to that of traditional organic solvents. NADES has a single component, and its application in extraction is greatly restricted, thus people pay more attention to the research of DES. DESs are liquids usually formed by hydrogen bond acceptors (HBA, salts such as quaternary ammonium salts) and hydrogen bond donors (HBD, such as polyols, urea, and carboxylic acids) [15,16]. As an extraction solvent, DES has the advantages of higher efficiency, shorter time, lower cost, no toxicity, biodegradability, and improved product purity [17], offering a wide range of application prospects [18,19,20,21]. Accordingly, determination of the utility of different DES types for extraction of active ingredients of Chinese herbal medicines has become a popular research direction. At present, DES-based methods have been extensively explored for abstraction of flavonoids from traditional Chinese medicines [22,23,24,25]. Zhao et al. [26] used choline chloride (ChCl)-ethylene glycol (Eg) (1:4) DES to extract rutin from *Sophora japonica* under the condition of 20% water content, and the extraction rate was as high as 194.17 ± 2.31 mg·g^−1^. Compared with 60% ethanol and 60% methanol, it has obvious advantages. Duan et al. [27] prepared betaine-malic acid (Ma) (1:1), L-proline-levulinic acid (La) (1:2), and choline chloride-N,N’-dimethylurea (1:1) from 14 kinds of DESs. They were used to extract icariin, and the results all showed that the extraction rate was higher than that of the methanol method. In order to extract baicalin from *Scutellaria baicalensis Georgi*, Wang et al. [28] prepared choline chloride-lactic acid (LA) DES with a molar ratio of 1:2. The results showed that the extraction rate of baicalin with DES combined with ultra-high-pressure extraction was 116.8 mg·g^−1^, which was higher than that of 70% ethanol and heat reflux method. Until now, the research on DESs used to extract flavonoids from AS has not been explored.

In this study, an ultrasonic-assisted DES extraction method was introduced and, for the first time, tailor-made DESs were used to extract flavonoids from AS. The feasibility and effectiveness of eight DES in AS flavonoid separation were evaluated. The Box–Behnken design (BBD)-response surface method (RSM) was used to optimize extraction conditions. Furthermore, macroporous resins was used to recover total flavonoids from DESs extracts and the recovered DESs was evaluated on the recycling utility. The procedure developed in this study provides a solid theoretical foundation for efficient and rapid extraction of flavonoids from AS.

## 2. Results

### 2.1. Determination of Total Flavonoids Content

The total flavonoids content of the AS extract was calculated the calibration curve as follows: y = 0.0066x − 0.0391, R^2^ = 0.9942. The rutin reference solution displayed a good linear relationship within a concentration range of 0 to 1.5 mg·mL*^−^*^1^.

### 2.2. Physical and Chemical Properties of DESs

The physical and chemical properties of DESs, such as density, melting point, conductivity, acidity and alkalinity, viscosity, and polarity, depend on their composition [29]. Analysis of these properties should facilitate effective extraction of target compounds from natural materials. The presence of phenolic hydroxyl groups in flavonoids makes them acidic [30]. Consequently, flavonoids are more stable under acidic conditions and easily soluble in polar solvents [31]. Earlier findings suggest that the extraction efficiency of alcohol and amine-based DESs for flavonoids is higher than that of sugar and acid DESs [32,33]. Due to the higher viscosity of the latter group, it is not conducive to the flow of the medium. In the current study, three types of DESs were selected for extraction of AS total flavonoids, specifically choline chloride, glycerol (Gly), and organic acids. Among them, the choline chloride in choline chloride DESs acts as hydrogen bond acceptor, the glycerol in glycerol DESs acts as hydrogen bond donor, and a series of organic acids in organic acid DESs act as hydrogen bond acceptors. Choline chloride (C_5_H_14_ClNO) is a white crystal at room temperature with an electron-deficient center of a nitrogen positive ion and generally used as a hydrogen bond acceptor. Based on earlier reports [15,34], we used ChCl as the hydrogen bond acceptor and ethylene glycol, levulinic acid, and 1,4-butanediol (Buta) as hydrogen bond donors, and prepared DESs at a range of molar ratios. Glycerol, also known as glycerin (C_3_H_8_O_3_), is a viscous liquid at room temperature. Due to the presence of three hydroxyl groups, the molecule has good capacity as a hydrogen bond supplier and is generally used as a hydrogen bond donor. With reference to extensive earlier research, Gly was used as the basic part and La, lactic acid, and malic acid added as hydrogen bond acceptors. Due to the lower viscosity of Gly, the viscosity of this type of DES is expected to be lower. The organic acid, La, is in a translucent crystalline state at room temperature. Citric acid (Ca) is 2-hydroxypropane-1,2,3-tricarboxylic acid in liquid form. Owing to the presence of more than one carboxyl group, Ca shows strong ability to accept hydrogen bond. Therefore, Buta and Gly were selected as hydrogen bond donors for DES preparation. The pH and viscosity of individual DESs were measured at 30 °C (Table 1). Among the eight DESs under investigation, Gly DESs have stronger acidity, and DES-4 and DES-7 showed lowest viscosity [35]. Furthermore, although DES-6 has the strongest pH, but its viscosity is at a medium level; it also exhibits a strong extraction effect. In addition, DES-8 has low viscosity and strong acidity, but it still does not show good extraction ability. This may be because Ca, as a hydrogen bond acceptor, has a weaker ability to accept hydrogen bonds due to its lower polarity. The interaction force formed between Ca and Gly is small, and it is difficult to achieve effective extraction of target components. The collective data (Table 1 and Figure 1) clearly indicate that physical and chemical properties of DESs are important factors affecting extraction capacity, and it is not only affected by one property.

### 2.3. Selection of DESs

We conducted a pre-experiment on Soxhlet and Batch, and the results verified the conjecture that the extraction rates of both were lower than those of UA-DES method. Thus, we directly compared the effects of eight types of DES (water content of 30%, solid-liquid ratio of 1:20 g·mL^−1^) under ultrasonic heating (UA) (40 kHz; 500 W; 40 °C; 20 min). The results are shown in Figure 1. Overall, three DESs (DES-4, DES-6 and DES-7) had high extraction rates, while the extraction effects of DES-5 and DES-8 were extremely poor. In general, the flavonoid extraction rate of Gly DESs was higher than that of the other two types of DESs, and at a Gly and La molar ratio of 1:1, the extraction rate was the highest, yielding 16.377 ± 0.177 mg·g^−1^, which was significantly greater compared with other DES types (*p* < 0.0001). This finding may be attributable to the low viscosity of the DES system composed of Gly and La, which can lead to high flow and diffusion, promoting suspension and dispersion of AS powder in the solvent and increasing the contact area, which is beneficial for total flavonoid extraction. Accordingly, DES composed of Gly and La was selected as the optimal extractant and total flavonoids of AS were extracted with ultrasonic assistance. A single factor experiment was subsequently designed to optimize the extraction parameters.

### 2.4. Fourier-Transform Infrared Spectroscopy (FT-IR) Analysis of Gly-La DES

In this experiment, levulinic acid was used as the starting HBA combined with glycerol to form a hydrophilic DES [36], and weakly polar flavonoids were extracted. The establishment of hydrogen bonds between two compounds was responsible for the formation of eutectic mixtures. Comparison of the FT-IR spectra of glycerin, levulinic acid and optimal solvent (Gly-La) revealed overlapping spectra of glycerin, levulinic acid and DES, as shown in Figure 2. The FT-IR spectrum has two regions, specifically, fingerprint (1333–400 cm^−1^) and functional (4000–1333 cm^−1^) groups. The FT-IR spectrum of Gly-La showed a shift in the OH band of glycerol from 3281.62 cm^−1^ to 3368.28 cm^−1^ and a strong broad peak at 3500–3200 cm^−1^, and the intrinsic frequency of the hydroxyls appeared at 3700–3000 cm^−1^. Corresponding to OH vibration, indicating the formation of a large number of DES. For the hydrogen bond in the infrared spectrum of 2800–2950 cm^−1^, a wider band was observed, indicating the existence of a strong -OH-O^−^ interaction [37], where the formation of hydrogen bonds considers the combination of the carbonyl group and carboxyl group in La with the hydroxyl group in Gly. This meant that the peaks at 3368.28 cm^−1^ and 2936.82 cm^−1^ in FT-IR spectra (Gly-La) contained hydrogen bonds. The carbonyl band of levulinic acid expanded from 1700.00 cm^−1^ and transferred to the high wave value of 1703.69 cm^−1^ in Gly-La. The strong peak corresponded to the stretching vibration of carbon oxide in the polymer (mainly C = C and C = 0). FT-IR spectra of glycerol and levulinic acid contained absorption peaks at 1028.84 cm^−1^ and 1158.70 cm^−1^, respectively, that shifted to 1038.23 cm^−1^ and 1164.77 cm^−1^ in the spectrum of Gly-La, which was related to expansion and contraction of single bonds. Vibration (mainly C-C, C-O, C-N) is closely related, since hydrogen bond formation leads to changes in the bond electron density, subsequent frequency of the stretching vibration, and ultimately, the spectrum. The above results provide theoretical support for the efficient extraction of target components using the Gly-La DES system. In addition, during infrared spectrum detection, CO_2_ background was deducted from all samples, therefore the influence of CO_2_ on infrared spectrum detection results was no longer considered.

### 2.5. Selection of the Most Suitable Ultrasonic Power Level

At an ultrasonic power of 500 W, the flavonoid yield was the largest (Figure 3). Upon an increase in ultrasonic power from 200 W to 500 W, the flavonoid yield was increased. With greater ultrasonic power, the molecular vibration speed was faster and intermolecular interactions strengthened, which promoted dissolution of flavonoids. In addition, with increasing temperature, the viscosity of the eutectic solvent was decreased and AS powder was more fully dissolved and dispersed, leading to more efficient extraction of flavonoids [38]. However, under conditions of continual increase in ultrasonic power, the yield of flavonoids was significantly decreased. One possible explanation is that high-frequency vibrations affect the network structure formed within DESs, leading to instability, and a strong cavitation effect degrades flavonoids and other components, thereby reducing the ability to extract flavonoids from AS. Accordingly, the ultrasonic power was set as 500 W.

### 2.6. Determination of the Optimal HBA-HBD Ratio

Most DESs are in liquid state at room temperature and their melting points are lower than any other component of the solvent [39]. The melting point mainly depends on interactions between the hydrogen bond acceptor and donor. Increased force is associated with greater degree of damage to the crystal structure and lower melting point of DESs [40]. During the preparation process, the melting point is mainly affected by the molar ratio between the hydrogen bond acceptor and donor [41]. The appropriate molar ratio is correlated with lower melting point and larger operating temperature range during the extraction process, which is more conducive to transmission of the medium. Therefore, optimization of the HBA-HBD ratio is crucial to improve the extraction rate of total flavonoids. In this study, Gly and La were used to formulate eutectic solvents under fixed conditions at a range of molar ratios (2:1, 1.5:1, 1:1, 1:1.5, and 1:2). The extraction rates of AS total flavonoids were measured (Figure 3). As the ratio of Gly-La gradually decreased, the extraction rate of total flavonoids initially increased and subsequently decreased. The extraction rate was significantly higher at a molar ratio of 1:1 (16.255 ± 0.313 mg·g*^−^*^1^) relative to the other molar ratios tested (*p* < 0.0001). The reason for this finding may be that reduction in the Gly content leads to decreased viscosity and surface tension of the deep eutectic solvent, thereby improving the diffusion and mass transfer effect [42]. Moreover, La is acidic. The partial acidic environment is beneficial for extraction of flavonoids, and therefore, the extraction efficiency increases with decreasing HBA-HBD ratio. However, excessive acid or alkali induce significant changes in the physicochemical properties of DES, such as pH, polarity, viscosity, and density, which can reduce extraction efficiency. Based on the results, a Gly-La ratio of 1:1 was selected for follow-up experiments.

### 2.7. Selection of the Most Suitable Water Content

Due to the large number of network structures composed of hydrogen bond donors and acceptors, van der Waals forces and other electrostatic interactions, the majority of DESs have relatively high viscosity at room temperature [43], thereby affecting the intermolecular interactions of the extracted liquid. Optimization of the fluidity of DESs plays a key role in improving the extraction rate [44]. Viscosity is not only related to the HBA-HBD ratio, but also the water content of DESs [45]. Accordingly, we further focused on the influence of different water contents on DES extraction capability (Figure 3). Upon increasing the water content of DESs from 20% to 60%, the extraction rate of total flavonoids was significantly altered, initially showing an increase followed by a decrease, with peak levels recorded at a water content of 30%. One possibility is that the addition of a small amount of water reduces DES viscosity, but also maintains their structure [46], thereby increasing the contact area between the drug and extractant, which facilitates dissolution of the target component. In addition, according to the principle of “similar compatibility” [47], water as a strong polar solvent increases the dipole/polarizability of DES and is therefore closer to the polarity of flavonoids. However, the extended dilution of DES with water will result in the loss of existing hydrogen bonds. Since polar water molecules are very flexible and easy to form hydrogen bonds with other solvents, excess water molecules will form hydrogen bonds with the carbonyl and carboxyl of La, and will form hydrogen bonds with the hydroxyl of Gly, therefore the number of hydrogen bonds between La and Gly in DES will naturally reduce. In other words, the addition of excessive water causes competition with groups in the two solvents to form hydrogen bonds, resulting in the reduction or even disappearance of DES special structures [48], and ultimately weakens interactions between DESs and the target components. Therefore, in follow-up experiments, a DES system with a water content of 30% was selected for further optimization.

### 2.8. Determination of the Optimal Solid-Liquid Ratio

With the increase of solid-liquid ratio of DES, the extraction rate of AS total flavonoids initially increased and subsequently decreased. At an increased solid-liquid ratio of 1:20 g·mL*^−^*^1^, the extraction rate reached a peak value of 16.259 ± 0.257 mg·g*^−^*^1^. Under the condition of solid-liquid ratio of 1:20 g·mL*^−^*^1^, the value of flavonoid content is higher than that of other groups (*p* < 0.0001), as shown in Figure 3. This finding may be explained by the fact that as the material-liquid ratio gradually decreases, full contact between the material and solvent is achieved and the solution mass transfer driving force enhanced, which is beneficial for dissolution of flavonoids. At a ratio of >1:20 g·mL*^−^*^1^, simply increasing the solvent ratio leads to a decrease in total flavonoid yield, possibly since diluted solvent will increase the energy consumption of ultrasound, resulting in ineffective breakdown of cells. The effect of ultrasonic cavitation becomes less pronounced and, consequently, release of effective substances is reduced. At a solid-liquid ratio of 1:20 g·mL*^−^*^1^, not only is the extraction effect optimal, but also the experimental costs are saved. Therefore, 1:20 g·mL*^−^*^1^ was selected to further optimize the extraction process.

### 2.9. Selection of the Suitable Extraction Temperature

Temperature is an important factor affecting various physical and chemical properties and functions of DESs [49]. With increasing temperature, the conductivity of DESs is elevated, along with a decrease in the surface tension [50]. As shown in Figure 3, as the extraction temperature gradually increased to 50 °C, the extraction rate of total flavonoids reached a peak. Above this temperature, the extraction rate of total flavonoids showed a downward trend. This finding may be explained by the possibility that increase in temperature of the extraction medium speeds up molecular movement and reduces solvent viscosity, thereby increasing the solubility of the compound and energy of DESs, leading to enhanced mass transfer rate. However, excessive temperature also accelerates the dissolution of flavonoids and other active ingredients, ultimately resulting in decreased extraction rates. Therefore, an extraction temperature of 50 °C was selected for subsequent experiments.

### 2.10. Selection of the Optimal Extraction Time

Extraction time is another important parameter influencing the extraction rate of AS total flavonoids. In our experiments, when the ultrasound time was increased from 40 min to 70 min, the extraction rate of total flavonoids was significantly increased and became stable with extended time (Figure 3). A potential explanation for this result is that the dissolution rate of the target component increases with time, but when osmotic pressure of the total solution system reaches equilibrium, mass transfer equilibrium between the DES system and AS powder is also reached. Moreover, long-term ultrasound leads to instability of the dissolved flavonoids. In addition, the excessively long ultrasound time was associated with significantly increased extraction costs. Ultimately, 70 min was selected as the extraction time for further optimization.

In addition, we consider levulinic acid is more acidic, and CO_2_ is less acidic, thus CO_2_ in the air is generally difficult to actively dissolve in DES composed of levulinic acid. In addition, we believe that even if Gly-La DES absorbs trace amounts of CO_2_, this will not affect the results of its ability to extract traditional Chinese medicine. Therefore, the CO_2_ factor was excluded when designing the single factor test.

### 2.11. RSM Optimization Test Data and Analysis

#### 2.11.1. RSM Optimization Test

On the basis of the single factor test and the Box–Behnken design principle, a four-factor, three-level response surface test was designed to optimize the extraction process of AS total flavonoids. The water content of DES (A, mL), solid-liquid ratio (B, g·mL^−1^), extraction temperature (C, °C), and extraction time (D, min) were taken as independent variables, with the extraction rate of total flavonoids (Y, mg·g^−1^) as the response variable of 29 groups. The results are presented in Table 2. Using Design Expert 13 software to perform quadratic regression fitting on data in Table 3, a regression model was obtained as follows:

Y = 22.77 − 0.9834A + 0.4785B + 1.47C + 0.9283D − 0.0697AB − 0.0285AC + 1.39AD − 0.9268BC − 1.05BD + 0.2705CD − 1.41A^2^ − 0.6931B^2^ − 1.76C^2^ − 1.69D^2^

Analysis of variance of data collated in Table 2 was conducted (presented in Table 3). The model F value was 12.40 (*p* < 0.0001). The *p* value of the lack-of-fit item was not significant at 0.1293, indicating that the equation is similar to actual data, which supports the utility of the combination. The model was selected to design the experiment for optimizing the extraction conditions of total flavonoids. The multiple correlation coefficient R^2^ value was 0.9254, indicating that this model accurately reflects the correlations between the four factors and the total flavonoid extraction rate. The adjusted R^2^ (Radj^2^
*=* 0.8507) value shows that the generated model is reasonable, and that only 0.1% of the response changes are due to unpredictable variables. In addition, the model displays better predictive ability (Rpred^2^
*=* 0.6038). According to the significance standard, the effects of primary terms A and D, interaction term AD, and quadratic term A^2^ on the extraction rate of AS total flavonoids reached a significant level (*p* < 0.01), along with the interaction terms BC and BD and quadratic term B^2^ (*p* < 0.05). According to the F value, the order of the degree of influence of each factor on the extraction rate of total flavonoids was determined as: extraction temperature > water content > extraction time > solid-liquid ratio.

#### 2.11.2. Analysis of the Response Surface Plot

Specific software was applied to perform quadratic multivariate fitting for obtaining a three-dimensional curve diagram (above) and corresponding contour diagram (below), which intuitively reflect the trend and extent of the interactions between various factors and their effects on the total flavonoid extraction rate (Y), as shown in Figure 4. The surface inclination of the response surface was positively correlated with interactions between the two factors. A larger surface slope (the greater the slope of the surface, the steeper the slope) signifies that the response value is more sensitive to factor changes, i.e., greater influence of factors on extraction rate of total flavonoids, and a smoother surface slope is indicative of a smaller influence. Elliptical contour lines indicate significant interactions between two factors while round contour lines correspond to low significance [51].

The yield of AS flavonoids did not significantly change with changes in the water content and solid-liquid ratio of the system (Figure 4a). The contour line was nearly circular, indicating that interactions between the water content and solid-liquid ratio have no significant effect on AS flavonoid yield. From Figure 4b,f, it is evident that the yield of AS flavonoids increased first and then decreased with increase in water content-extraction temperature and extraction temperature-extraction time. However, the magnitude of the change was relatively small. The contour lines were round, signifying those interactions between water content and extraction temperature and those between extraction temperature and extraction time have no significant impact on AS flavonoid yield. In chart 4c, the extraction rate of total flavonoids increased with increasing water content and decreasing extraction time and the slope of the curved surface was relatively large. The contour line was elliptical, revealing that the mutual effects between the two have a remarkable impact on AS flavonoid yield. Response surface curves formed by extraction temperature/solid-liquid ratio and extraction time/solid-liquid ratio were steep, as shown in Figure 4d,e. Simultaneously, their contour lines were elliptical and relatively tight, indicating significant interactions between extraction temperature/solid-liquid ratio and extraction time/solid-liquid ratio. Notably, these findings correspond to analysis of variance data in Table 3.

#### 2.11.3. Model Verification and Method Comparison

According to the model equation fitting prediction combined with the above single factor test results (Gly:La *=* 1:1, ultrasonic power 500 W), the optimal extraction conditions for total flavonoids were as follows: water content 27.928%, solid-liquid ratio 1:18.103 g·mL*^−^*^1^, extraction temperature 54.902 °C, and extraction time 72.869 min. Under these conditions, the extraction rate of total flavonoids was predicted to be 23.318 mg·g*^−^*^1^. Taking into account the actual operation, the conditions were adjusted as follows: water content 28%, solid-liquid ratio 1:18 g·mL*^−^*^1^, extraction temperature 55 °C, and extraction time 73 min. Parallel extraction under these conditions was performed three times. The verification test revealed an average extraction rate of AS total flavonoids of 23.926 ± 0.128 mg·g*^−^*^1^, which is extremely close to the predicted value, and a relative error of less than 2.62%. At the same time, the extraction rate of AS flavonoids extracted by the same power ultrasonic-assisted ethanol method (ethanol concentration 75%, solid-liquid ratio 1:18 g·mL*^−^*^1^, extraction temperature 55 °C, and extraction time 73 min) was tested. Our results showed that the extraction rate using the DES method was 40.7% higher than ultrasonic-assisted ethanol extraction method (17.009 ± 0.216 mg·g*^−^*^1^) (*p* < 0.0001), clearly demonstrating highly improved extraction potential.

### 2.12. SEM Observation of AS Powder Microstructure

We further explored the relationship between the degrees of rupture of plant cells with different extraction methods. To this end, the microstructure of AS powder before and after treatment was examined via SEM (Figure 5, ×2000 magnification). Cells of the powdered form before extraction (Figure 5a) were intact, the cell wall surface was mostly smooth, and adjacent pores of the cell wall were closely arranged. The texture structure was clearly detectable with no trace of damage. However, all extraction methods induce a certain degree of damage to AS cells. After ultrasonic-assisted ethanol treatment (Figure 5b), the cell wall of the sample became rough and small ridges and wrinkles appeared on the tissue surface. Following application of ultrasonic-assisted DES treatment (Figure 5c), cells showed complete rupture and collapse. The tissue surface was obviously damaged, and we observed the appearance of pores. Adjacent pores are loosely arranged, and more irregular fragments were evident. These findings clearly indicate that DESs breaks down cellulose in the cell wall [52]. Accordingly, we propose the degree of rupture of AS cells is positively correlated with the extraction rate of total flavonoids, consistent with the measurement result of extraction rate. In addition, the effects of different screens (40, 60, 80, 100 mesh) on the extraction of flavonoids from AS powder by UA-DES were preliminarily studied. The results showed that the extraction rate at 40 mesh was slightly lower, and there was almost no difference in the extraction rates of other items. In addition, if the particle size of the powder is too small, it is not convenient to separate it from the crude extract, therefore 60 mesh Acanthopanax powder was selected for follow-up test. In addition, SEM has a large scanning range, thus, in principle, from 1 nm to mm can be used for particle size analysis. We consider that the size of the sample particles does not affect the fine structures observed under the SEM because the structures presented are small enough.

### 2.13. Recovery of Total Flavonoids and Reusability of DES

The AS flavonoids must be effectively recovered and purified from DES extraction solvent. The documented recovery methods include macroporous resin, solid phase extraction [53], and supercritical carbon dioxide [54]. In the current research, the macroporous resin enrichment method was used to recover total flavonoids from DES extraction solvent. The macroporous resin enrichment method is simple to operate and low in cost. However, limited research date has evaluated its utility in recovery of AS flavonoids. Upon enrichment of the target component with organic extraction solution, it is necessary to remove the highly concentrated organic solvent and dilute it with water before further use in the resin adsorption and desorption steps. DESs which has been diluted with water can be directly used for resin enrichment. The hydrogen bond in DES is an important factor in target component extraction and the water content affects hydrogen bond formation between HBA and HBD. In addition, viscosity of DESs is relatively high, which leads to certain degree of effect on adsorption during the enrichment process. To achieve high-efficiency adsorption of flavonoids by the resin, earlier literature was used as a reference [55] in order to dilute the crude DES flavonoids solution to the certain concentration for subsequent experiments. Macroporous resin enrichment of total flavonoids from the extract and recovery of DES experiments were carried out at room temperature.

#### 2.13.1. Screening for the Resin

In this study, the resin was selected by comparing total recovery rates of AS flavonoids in DESs using four macroporous resin types. Adsorption of macroporous resin is affected by its physical properties [56]. For example, larger specific surface area and average pore size are associated with stronger adsorption. When the polarity of the resin is similar to that of adsorbate, high recovery capacity is achieved. This involves the “like solvent like” principle [57]. The physical properties of the four resins and recovery data are presented in are shown in Table 4. AB-8 macroporous resin had the best recovery effect on total flavonoids (68.95 ± 0.074%), followed by HPD600 (56.54 ± 0.066%), which was slightly higher than S-8 (55.85 ± 0.181%) and D101 resin (55.76 ± 0.222%). The total adsorption rate of AB-8 resin (77.10 ± 0.254%) was significantly higher (*p* < 0.0001) compared with the other three resins.

#### 2.13.2. Screening Eluent Solvent

The desorption solvent is related to polarity [58], and depends on the solubility of flavonoids. The organic solvents, such as methanol, ethanol, ethyl acetate, etc., were used for the desorption of flavonoids from resin. In this study, the desorption solvent was selected from methanol, acetic acid, ethyl acetate, 50% ethanol, 75% ethanol, and absolute ethanol, as shown in Figure 6. We observed an optimal desorption effect at 75% ethanol. The total flavonoid desorption rate was 92.32 ± 0.158%. Therefore, 75% ethanol was chosen as the optimal desorption solvent. Under optimal conditions, the recovery rate of AS total flavonoids was 71.56 ± 0.256% (46.8 ± 0.515% purity), which was significantly higher relative to that with other desorption solvents (*p* < 0.0001).

#### 2.13.3. Reusability of DES

Novel high-efficiency DES extraction agents can be recovered/recycled. After evaporation of water from the recovered DES in vacuum, a new extraction solvent was prepared based on the appropriate water content and further used to extract total flavonoids from AS. As shown in Figure 7, under optimized conditions, with increasing number of reuses of recovered DES, the content to extract target components was decreased. The extraction rates for DES upon the second and third rounds were 20.172 ± 0.487 mg·g^−1^ and 10.005 ± 0.366 mg·g^−1^, and extraction efficiencies were 84.23% and 41.78%, respectively. Our results demonstrate that DESs can be reused at least twice, leading to significant cost saving.

## 3. Materials and Methods

### 3.1. Reagents and Instruments

*A. senticosus* was purchased from Liaocheng Limin Pharmacy (Liaocheng, China), ground into powder with a grinder (H8422, Hebei Huicai Technology Co. Ltd., Hebei, China), passed through a 60-mesh sieve, and stored at room temperature.

Choline chloride (AR, 98%), ethylene glycol (AR, 99.5%), levulinic acid (AR, 99%), 1,4-butanediol (AR, 98%), and lactic acid (ACS, ≥85%) were purchased from Shanghai Aladdin Biochemical Technology Co., Ltd. (Shanghai, China). Aluminum nitrate (AR, 99%), citric acid (AR, ≥99.5%), glycerin (AR, 99%) and DL-malic acid (BR) were acquired from Shanghai Macklin Biochemical Co., Ltd. (Shanghai, China). Ethanol (AR, 95%), sodium hydroxide (AR, 95%), Methanol (AR, ≥99.9%), glacial acetic acid (AR, ≥99.5%), sodium nitrite (AR, ≥99.0%), and ethyl acetate (AR, ≥99.5%) were obtained from Yantai Far East Fine Chemical Co., Ltd. (Shandong, China). The four macroporous resins, D-101, HPD-600, S-8, and AB-8, were obtained from Donghong Chemical Co., Ltd. (Shanxi, China). Deionized water was prepared using a laboratory ultrapure water machine (model DBW-UP-10; Dongguan Dongbo Water Treatment Co., Ltd., Guangdong, China). All other chemicals were at least of analytical grade.

The following instruments were employed: viscometer (model Ultra DV-III; American Brookfield Co., Ltd., Middleboro, MA, USA), Fourier transform infrared spectrometer (model Thermo Nicolet iS5, Zequan International Group Shanghai Zequan Instrument Equipment Co., Ltd., Shanghai, China), pH meter (model P811; Shanghai Youke Instrument Co., Ltd., Shanghai, China), UV-visible spectrophotometer (model UV-5900; Shanghai Yuanxi Instrument Co., Ltd., Shanghai, China), centrifuges (model H1065-W; Hunan Xiangyi Laboratory Instrument Development Co., Ltd., Changsha, China), ultrasonic bath (Power-Sonic SB-600DTY; Ningbo Xinzhi Bio Technology Co., Ltd., Ningbo, China), constant temperature electric water tank (model DK-8D; Shanghai Yiheng Technology Co., Ltd., Shanghai, China), constant temperature oscillator (model HZQ-F160A; Shanghai Yiheng Scientific Instrument Co., Ltd., Shanghai, China), scanning electron microscope (model SU8010, Changsha Kemei Analytical Instrument Co., Ltd., Changsha, China), and peristaltic pump driver (model BT100-2J; Baoding Lange Constant Current Pump Co., Ltd., Baoding, China).

### 3.2. Experimental Methods

#### 3.2.1. Preparation of DESs

HBA and HBD reagents were mixed at an appropriate molar ratio, and heated in a water bath at 80 °C with constant agitation until a clear and homogeneous liquid was formed [59]. According to prediction test, Choline chloride, levulinic acid, lactic acid, malic acid, and citric acid were selected as HBA and ethylene glycol, La, 1,4-butanediol, and glycerol as HBD. Molar ratio was showed in Table 1, and eight different types of DESs were prepared.

#### 3.2.2. Determination of pH and Viscosity of DES

The pH value of DES solution with 30% water content was measured using a pH meter at a set temperature of 30 °C. The viscosity of DES solution was measured using the Brookfield DV-III rotary viscometer. Each DES was measured in triplicate.

#### 3.2.3. Determination of Total Flavonoids Content

The content of total flavonoids was determined using the NaNO_2_-Al(NO_3_)_3_-NaOH colorimetric method described by Feng et al. [60] with slight modifications. The rutin was used as the reference. In brief, the rutin was dissolved in DES to obtain a final concentration of 1.5 mg·mL^−1^, which was used to prepare standard solutions (0, 0.25, 0.5, 0.75, 1.0, 1.25, and 1.5 mg·mL^−1^).1 mL standard solutions or extract was accurately placed in a 25 mL volumetric flask, then, 1 mL of 5% (m/v) NaNO_2_ solution added, and shaken for 6 min. Subsequently, addition of 1 mL of 10% (m/v) Al(NO_3_)_3_, mixed and incubated for 6 min. Finally, 9 mL of 4% NaOH solution was added, added 70% ethanol up to 25 mL, with 15 min standing. Absorbance was measured at 510 nm to calculate the total flavonoid content.

#### 3.2.4. Extraction Process

DES (7 mL) configured according to the corresponding molar ratio (Table 1) was added to a 20 mL centrifuge tube, followed by 3 mL water (DES water content was 30%). The solution was heated and thoroughly mixed in a water bath at 80 °C to prepare DES extraction solvent with a solid-liquid ratio of 1:20 g·mL^−1^. AS powder (0.5 g) was accurately weighed and placed in DES. Extraction rates under mechanical shaking with constant temperature oscillator-assisted extraction (200, 225 or 250 r·min^−1^, 40 min, 40 °C) and ultrasonic-assisted extraction (500 W, 40 min, 40 °C) were compared.

#### 3.2.5. Single Factor Effects

The effects of six single factors, specifically, Ultrasonic power (200, 300, 400, 500, and 600 W), HBA:HBD molar ratio (2:1, 1.5:1, 1:1, 1:1.5, and 1:2), DES water content (20%, 30%, 40%, 50%, and 60%), solid–liquid ratio of AS powder to DES solvent volume (1:10, 1:20, 1:30, 1:40, and 1:50 g·mL^−1^), extraction temperature (30, 40, 50, 60, and 70 °C), and extraction time (40, 50, 60, 70, and 80 min), were evaluated on extraction yields of flavonoids.

#### 3.2.6. Optimization Extraction Method with RSM

BBD was conducted by selecting the optimal levels each of the four single factors (DES water content, solid-liquid ratio, extraction temperature, and extraction time). On the basis of the experimental data, three levels of response surface experiments were conducted for each factor (Table 5), and the extraction rate of total flavonoid used as an evaluation index to analyze optimal combinations of the four factors.

#### 3.2.7. Fourier-Infrared Spectrophotometry

FT-IR spectral measurements were performed using a Thermo Nicolet iS5 instrument. Samples were placed in the liquid pool for testing. Spectral data were recorded between 4000 and 400 cm^−1^ at room temperature.

#### 3.2.8. SEM Observation of AS Powder Microstructure

Scanning electron microscopy (SEM) is commonly used to determine the surface morphology and structure of materials. The surface of the sample is scanned using electron beams to obtain three-dimensional spatial information [61]. The structural changes in AS powder with ultrasonic-assisted ethanol extraction were compared in this study. After drying, AS powder was adhered to the conductive glue thinly and evenly and installed on the aluminum post. After spraying a gold coating, imaging with SEM was performed at an acceleration voltage of 10 kV and 2000 × amplification.

#### 3.2.9. Total Flavonoid Recovery Using Macroporous Resin and Reusability of DES

I. Macroporous resin pretreatment

Macroporous resin pretreatment was performed according to a previous report [62], with appropriate adjustments. Specifically, macroporous resin was soaked with absolute ethanol for 24 h and washed with deionized water until the alcohol was no longer present. The resin was further soaked in five times the volume of 5% (m/v) NaOH for 6 h and washed with deionized water to ensure neutrality, followed by soaking in five times the volume of 5% (*v*/*v*) HCL for 4–5 h, and re-washing until a neutral pH was achieved. For storage, the resin was soaked in absolute ethanol and washed again until the alcohol was completely removed for later assay.

II. Screening for the optimal macroporous resin and desorption solvent

Columns (16 mm × 40 cm) were wet packed with the pretreated macroporous resin. DES extract solution of flavonoids was diluted with water and loaded the column with a flow rate of 1 mL·min^−1^. The column was eluted with deionized water at the same speed until the ethanol was completely washed off, followed by 70% ethanol at a flow rate of 1 mL·min^−1^ for dynamic desorption. All experiments were performed in triplicate. The total flavonoid recovery rate (*W_1_*, %) were calculated using followed formula [63]:$$\mathrm{Total}\;\mathrm{flavonoid}\;\mathrm{recovery}\;\mathrm{rate}\text{:}\;W_{1}=\frac{C_{3}\times V_{2}}{C_{1}\times V_{1}}\times 100\%$$
where *C_1_* represents the mass of flavonoids before adsorption (mg·mL^−1^), *C_2_* the mass concentration of flavonoids after adsorption (mg·mL^−1^), *C_3_* the mass concentration of flavonoids after desorption (mg·mL^−1^), *V_1_* the volume of sample liquid (mL), and *V_2_* the volume of desorption liquid (mL).

III. Reusability of DES

The recovered DES was evaporated for remove water and reused under optimized conditions. All experiments were repeated three times in parallel according to the above steps and the total flavonoid yield calculated each time to establish the reusability of DES.

### 3.3. Statistical Analysis

SPSS statistical software (Version 13.0) was used to analyze data, which were expressed as mean ±SD and examined using one-way ANOVA.

## 4. Conclusions

In present study, an eco-friendly extraction method was established and optimized to extract flavonoids from AS. The DES system composed of a Gly-La molar ratio of 1:1 achieved the best extraction effect. Under 500 W ultrasonic power, the optimal conditions (water content of 28%, solid-liquid ratio of 1:18 g·mL^−1^, extraction temperature of 55 °C, and extraction time of 73 min), up to 23.928 ± 0.071 mg·g^−1^ flavonoid of AS was extracted, which was significantly higher compared to traditional ultrasonic assisted ethanol extraction. The macroporous resin AB-8 had high efficiency of recovered flavonoids from DES extraction liquid. Moreover, the DES can be reused twice at least. Therefore, our findings not only validate the effectiveness of DESs as a green, economical, and sustainable natural medium for extraction of flavonoids in traditional Chinese medicine, but also provide a strong theoretical foundation for their practical applications.

## Figures and Tables

**Figure 1 molecules-27-00923-f001:**
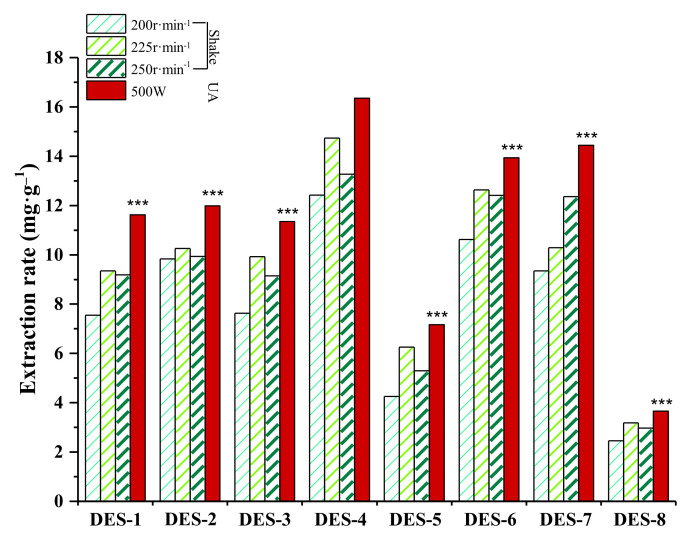
Effects of auxiliary operation on the extraction rates of total AS flavonoids. Note: Values represent mean ±SD of three independent experiments (compared with the extraction rate of DES-4; *** *p* < 0.0001).

**Figure 2 molecules-27-00923-f002:**
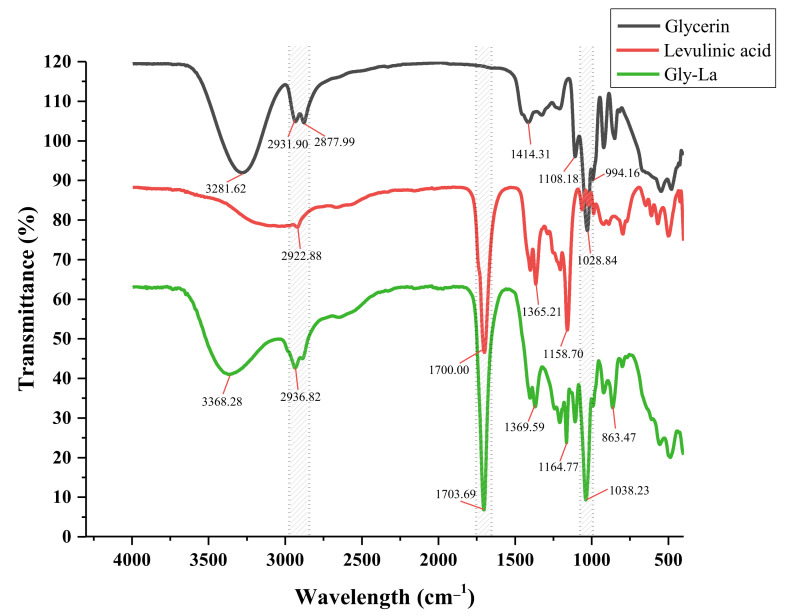
FT-IR spectra of glycerin, levulinic acid and Gly-La.

**Figure 3 molecules-27-00923-f003:**
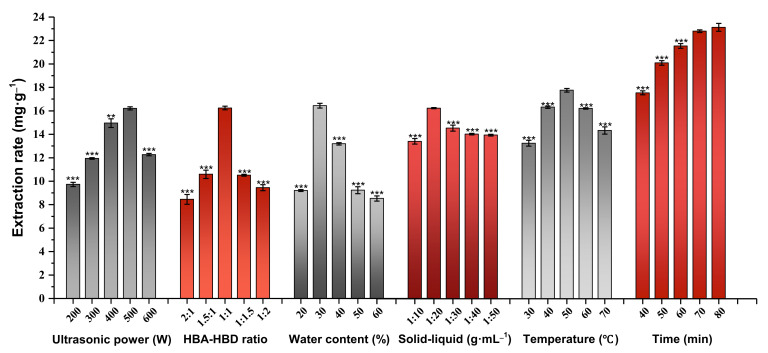
Effects of ultrasonic power (W), HBA-HBD ratio, water content (%), solid-liquid ratio (g·mL^−1^), extraction temperature (°C), and extraction time (min) on the extraction rate of total flavonoids from AS. Note: Values represent mean ±SD of three independent experiments (compared with the highest extraction rate under optimal conditions in each group; ** *p* < 0.01 and *** *p* < 0.0001).

**Figure 4 molecules-27-00923-f004:**
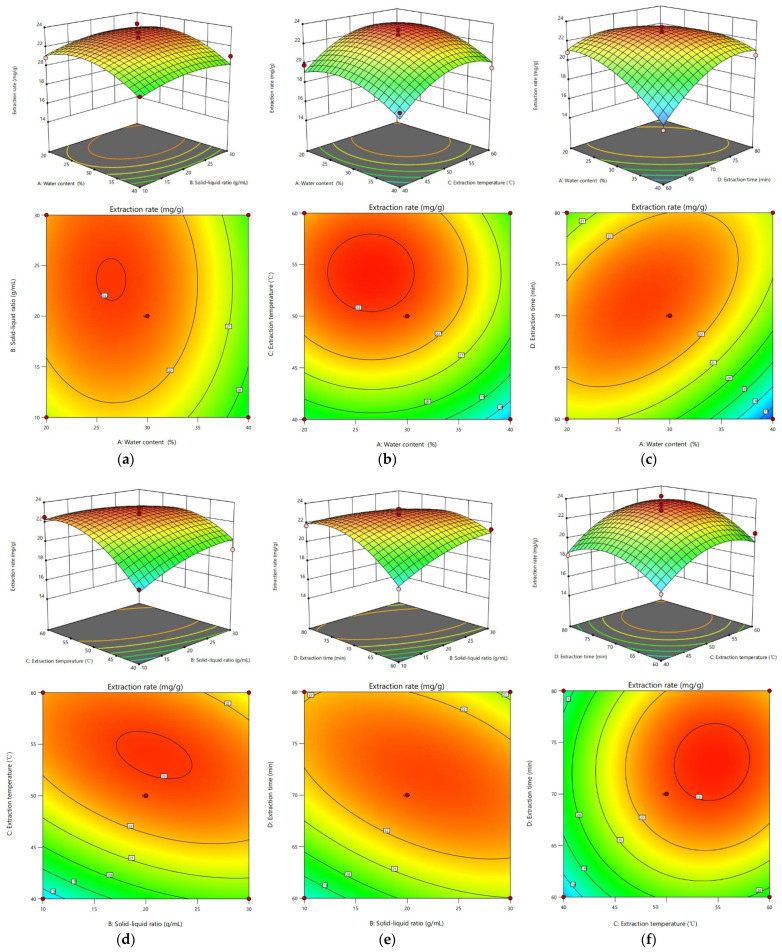
3D response surface plots and contour plots (**a**–**f**) of the influence of interactions of various factors on the extraction rate of AS total flavonoids.

**Figure 5 molecules-27-00923-f005:**
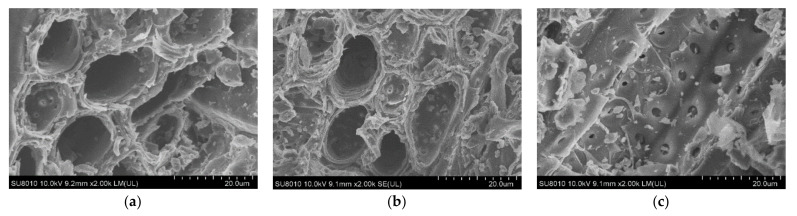
SEM images of AS powder (**a**) before processing and after extraction with (**b**) ultrasonic-assisted ethanol and (**c**) ultrasonic-assisted DES.

**Figure 6 molecules-27-00923-f006:**
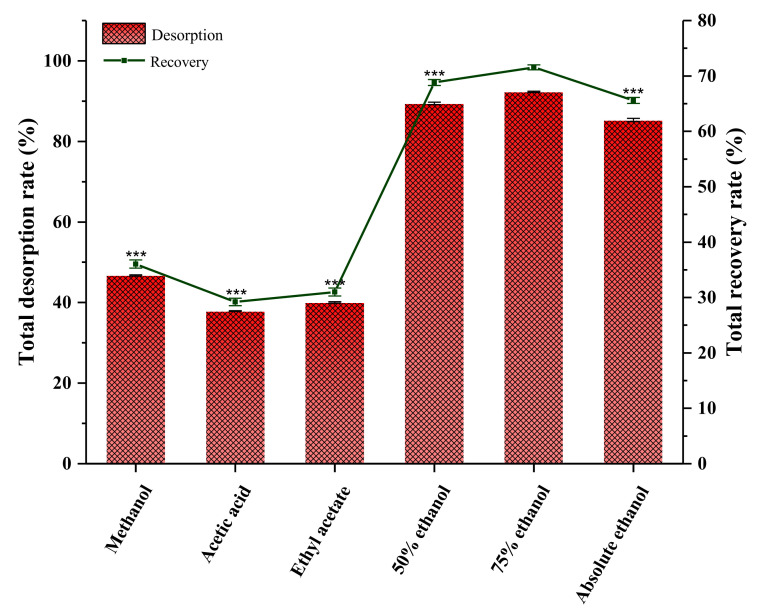
Comparison of total desorption and total recovery rates of AS total flavonoids with different elution solvents. Note: Values represent mean ±SD of three independent experiments (compared with the total recovery rate of 75% ethanol; *** *p* < 0.0001).

**Figure 7 molecules-27-00923-f007:**
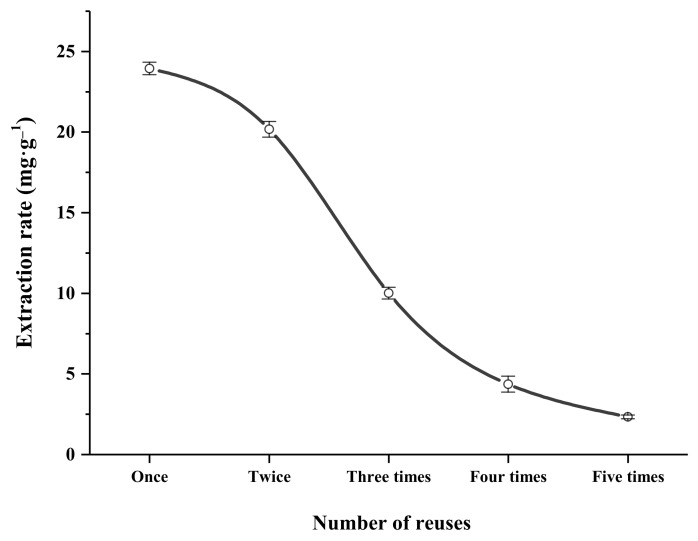
Reusability of recycled DES.

**Table 1 molecules-27-00923-t001:** Physical and chemical properties of deep eutectic solvents.

Serial Number	HBA	HBD	DESs		
			Molar Ratio	pH	Viscosity/mPa∙s
DES-1	ChCl	Eg	1:2	4.40	26.00 119.00 47.00 12.00 21.00 45.39 9.65 64.01
DES-2	ChCl	La	1:2	1.40	
DES-3	ChCl	Buta	1:3	3.74	
DES-4	La	Gly	1:1	1.85	
DES-5	LA	Gly	1:1	1.86	
DES-6	Ma	Gly	1:1	1.03	
DES-7	La	Buta	2:1	2.25	
DES-8	Ca	Gly	1:2	1.06	

**Table 2 molecules-27-00923-t002:** Response surface optimization of investigated variables using DES-4 as extraction solvent.

Run	A Water Content (%)	B Solid-Liquid Ratio (g/mL)	C Extraction Temperature (°C)	D Extraction Time (min)	Extraction Rate (mg/g)
1	0	0	−1	−1	16.953
2	−1	0	−1	0	19.815
3	1	1	0	0	20.963
4	0	−1	1	0	22.500
5	−1	0	1	0	21.647
6	−1	−1	0	0	20.85
7	0	0	1	−1	20.502
8	0	0	0	0	22.089
9	0	0	0	0	22.667
10	1	0	−1	0	17.754
11	1	0	0	1	20.502
12	0	1	−1	0	19.202
13	0	−1	0	−1	17.700
14	−1	0	0	1	19.929
15	0	−1	0	1	21.700
16	1	0	0	−1	15.863
17	0	1	0	−1	21.317
18	1	0	1	0	19.472
19	0	0	−1	1	18.212
20	0	−1	−1	0	17.694
21	0	0	0	0	22.844
22	0	0	0	0	22.817
23	0	0	0	0	23.418
24	0	1	1	0	20.301
25	−1	1	0	0	22.68
26	−1	0	0	−1	20.846
27	0	0	1	1	22.843
28	0	1	0	1	21.135
29	1	−1	0	0	19.412

**Table 3 molecules-27-00923-t003:** ANOVA of the regression model for extraction efficiency of flavonoids. Note: Values represent mean ±SD of three independent experiments (compared with the highest extraction rate under optimal conditions in each group; * *p*< 0.05 ** *p* < 0.01 and *** *p* < 0.0001).

Source	Sum of Squares	df	Mean Square	F-Value	*p*-Value	Significance
Model	104.04	14	7.43	12.4	<0.0001	significant
A	11.61	1	11.61	19.36	0.0006	**
B	2.75	1	2.75	4.58	0.0504	
C	25.92	1	25.92	43.24	<0.0001	***
D	10.34	1	10.34	17.25	0.001	**
AB	0.0195	1	0.0195	0.0325	0.8596	
AC	0.0032	1	0.0032	0.0054	0.9424	
AD	7.72	1	7.72	12.88	0.003	**
BC	3.44	1	3.44	5.73	0.0312	*
BD	4.37	1	4.37	7.29	0.0172	*
CD	0.2927	1	0.2927	0.4883	0.4961	
A^2^	12.86	1	12.86	21.46	0.0004	**
B^2^	3.12	1	3.12	5.2	0.0388	*
C^2^	20.16	1	20.16	33.63	<0.0001	***
D^2^	18.46	1	18.46	30.8	<0.0001	***
Residual	8.39	14	0.5994			
Lack of Fit	7.49	10	0.749	3.32	0.1293	not significant
Pure Error	0.9019	4	0.2255			
Cor Total	112.43	28				
R-Squared	0.9254					
Adj R-Squared	0.8507					
Pred R-Squared	0.6038					
Adeq Precision	11.4871					

**Table 4 molecules-27-00923-t004:** Physical properties, total adsorption capacity and total desorption capacity of four resins at 303 K.

Resin Types	Polarity	Specific Surface Area (m^2^·g ^−1^)	Average Pore Diameter/A°	Total Adsorption Rate/%	Total Desorption Rate/%	Recovery Rate/%
D101	Non-polar	500–550	90–100	66.46 ± 0.389 ^bc^	83.77 ± 0.237 ^c^	55.76 ± 0.222 ^cd^
AB-8	Weakly polar	480–520	130–140	77.10 ± 0.254 ^a^	89.38 ± 0.143 ^a^	77.10 ± 0.254 ^a^
S-8	Polar	100–120	280–300	67.90 ± 0.111 ^b^	82.25 ± 0.150 ^d^	55.85 ± 0.181 ^c^
HPD-600	Polar	550–600	80	66.13 ± 0.085 ^cd^	85.47 ± 0.168 ^b^	56.54 ± 0.066 ^b^

Note: Different letters of superscript in the same column indicate significant differences (*p* < 0.0001).

**Table 5 molecules-27-00923-t005:** BBD experimental design levels.

Levels	−1	0	1
Water content (%)	20	30	40
Solid-liquid ratio (g·mL^−1^)	10	20	30
Extraction temperature (°C)	40	50	60
Extraction time (min)	60	70	80

## Data Availability

The study did not report any data.

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
