# Peer review of "A Green Method of Extracting and Recovering Flavonoids from Acanthopanax senticosus Using Deep Eutectic Solvents"

_molecules, 2022, doi:10.3390/molecules27030923_

Round 1

Reviewer 1 Report

Manuscript ID: molecules-1514533

Title: A green method of extracting and recovering flavonoids from Acanthopanax senticosus using deep eutectic solvents

Comment 1: Compare the chemical (incl. flavonoids) and biological similarities between extracts obtained from the proposed method from Siberian ginseng (Acanthopanax senticosus) and Phak paem (Acanthopanax trifoliatus). (The somatic embryogenesis must be included in your response)

Comment 2: "In this study, the efficiencies of eight DESs in extracting flavonoids from Acanthopanax senticosus (AS) were compared" The effect of DESs on AS viability, behavior, and histology must be studied.

Comment 3: "In the current study, three types of DESs were selected for extraction of AS total flavonoids, specifically, choline chloride, glycerol (Gly) and organic acids" Clarify the hydrogen bond donor-acceptor cosolvent with DESs.

Comment 4: "Fourier-Transform Infrared Spectroscopy (FT-IR) analysis of Gly-La DES" Show the hydrogen tunneling above room temperature. What is the intrinsic frequency of the hydroxyls, that is, if these groups are not involved in any H-bonding? And what reference ν3 frequency should be used in the IR spectral during CO2 adsorption?

Comment 5: Fig.1, Table 1, and "However, excessive water induces a decrease in hydrogen bond acceptance alkalinity, destroys the hydrogen bond network structure, and ultimately weakens interactions between DESs and the target components" The relations between weak acidic groups and DESs in application and their affecting roles in extracting and recovering flavonoids must be explained. In addition, the CO2 effect on the solubility of flavonoids and solvent viscosity must be screened and graphed.

Comment 6: "These solvents usually contain secondary metabolites, such as sugar alcohols, sugars, amino acids and organic acids, as NADES components" Which is the most proper solvent: IL, DES or NADES for AS?

Comment 7: "When the polarity of the resin is similar to that of adsorbate, high recovery capacity is achieved" Consider the first principles polarizable force field.

Comment 8: "SEM observation of AS powder microstructure" Consider effect of the particle size of the powder in the discussion.

Comment 9: "2.13. Recovery of total flavonoids and reusability of DES" Add the temperature and %purity.

Comment 10: Figs.1,3,6 and 7 – Compare the current findings with UA-DES, Soxhlet and batch.

Comment 11: "One possibility is that the addition of a small amount of water reduces DES viscosity, thereby increasing the contact area between the drug and extractant, which facilitates dissolution of the target component" Evaluate the viscosity (and permeation) effects on drug dissolution and link the uptakes quantity of water to the increase of shear viscosity.

Comment 12: "In addition, according to the principle of “similar compatibility”, water as a strong polar solvent increases the dipole/polarizability of DES and is therefore closer to the polarity of flavonoids" The theoretical calculations of the concentrations of solvent-water hydrogen-bonded species and application to thermosolvatochromism of dipole/polarizability of DES are requested.

Comment 13: "Choline chloride (C5H14ClNO) is a white crystal at room temperature with an electron-deficient center of a nitrogen positive ion and generally used as a hydrogen acceptor" and "Vibration (mainly C-C, C-O, C-N) is closely related, since hydrogen bond formation leads to changes in the bond electron density, subsequent frequency of the stretching vibration, and ultimately, the spectrum" Study the influence of solvent viscosity, polarity and polarizability on the chemiluminescence parameters of inter and intramolecular electron transfer initiated chemiexcitation quantum yield systems. (Use computational chemistry to explore experimental solvent parameters)

Comment 14: "Due to the large number of network structures composed of hydrogen bond donors and acceptors, van der Waals forces and other electrostatic interactions, the majority of DESs have relatively high viscosity at room temperature[32], thereby affecting the intermolecular interactions of the extracted liquid" Evaluate the electrostatic response underlying the 3D-RISM theory (with the bridge diagrams) and its general relationship to the suggested models.

Comment 15: The language should be revised.

Comment 16: References:

Note 1: "Acanthopanax senticosus, also known as Siberian ginseng in China, is widely distributed in Russia and Asia" Add the following reference:

  1. Anti-tumour effects of polysaccharide extracted from Acanthopanax senticosus and cell-mediated immunity. Experimental and Therapeutic Medicine.

Note 2: "Among them, ILs are composed of multiple components with characteristic ionic bonds formed between cations and anions" and "However, one notable limitation is that many ILs are based on imidazole and pyridine and therefore exert toxic effects." Add the following reference:

2020.Ionic Liquids Toxicity—Benefits and Threats. International Journal of Molecular Sciences 21(17), 6267.

Note 3: "The physical and chemical properties of DESs, such as density, melting point, conductivity, acidity and alkalinity, viscosity and polarity, depend on their composition" Add the following reference:

2020.Overview of acidic deep eutectic solvents on synthesis, properties and applications. Green Energy & Environment 5(1), 8-12.

Note 4: "The presence of phenolic hydroxyl groups in flavonoids makes them acidic" Add the following reference:

  1. Chemistry and Biological Activities of Flavonoids: An Overview. The Scientific World Journal 2013. 1-16.

Note 5: "Consequently, flavonoids are more stable under acidic conditions and easily soluble in polar solvents" Add the following reference:

2020.Extraction of Flavonoids from Scutellariae Radix using Ultrasound-Assisted Deep Eutectic Solvents and Evaluation of Their Anti-Inflammatory Activities. ACS Omega 5(36), 23140–23147.

Note 6: "Among the DESs under investigation, Gly DESs have stronger acidity, and DES-4 and DES-7 showed lowest viscosity" Add the following reference:

  1. Viscosity model for choline chloride-based deep eutectic solvents. Asia-Pacific Journal of Chemical Engineering 10(2), 273-281.

Note 7: "For the hydrogen bond in the infrared spectrum of 2800–2950 cm−1, a wider band was observed, indicating the existence of a strong -OH-Cl− interaction." Add the following reference:

  1. Developed greener method based on MW implementation in manufacturing CNFs. International Journal of Nanomanufacturing 15(3), 269-289.

Note 8: "In addition, with increasing temperature, the viscosity of the eutectic solvent was decreased and AS powder was more fully dissolved and dispersed, leading to more efficient extraction of flavonoids" Add the following reference:

2021.Deep Eutectic Solvents for the Extraction of Bioactive Compounds from Natural Sources and Agricultural By-Products. Applied Sciences 11(11), 4897.

Note 9: "Increased force is associated with greater degree of damage to the crystal structure and lower melting point of DESs" Add the following reference:

2019.Deep Eutectic Solvents for Pretreatment, Extraction, and Catalysis of Biomass and Food Waste. Molecules 24(22), 4012.

Note 10: "During the preparation process, the melting point is mainly affected by the molar ratio between the hydrogen bond acceptor and donor" Add the following reference:

2020.Study on the Dissolution Mechanism of Cellulose by ChCl-Based Deep Eutectic Solvents. Materials 13(2), 278.

Note 11: "Optimization of the fluidity of DESs plays a key role in improving the extraction rate" Add the following reference:

2021.Optimization and kinetic study of ultrasound assisted deep eutectic solvent based extraction: A greener route for extraction of curcuminoids from Curcuma longa. Ultrasonics Sonochemistry 70, 105267.

Note 12: "Viscosity is not only related to the HBA-HBD ratio but also the water content of DESs" Add the following reference:

2021.Insights on the water effect on deep eutectic solvents properties and structuring: The archetypical case of choline chloride + ethylene glycol. Journal of Molecular Liquids 344, 117717.

Note 13: "Temperature is an important factor affecting various physical and chemical propeties and functions of DESs" Add the following reference:

2017.The study on temperature dependence of viscosity and surface tension of several Phosphonium-based deep eutectic solvents. Journal of Molecular Liquids 241, 500-510.

Note 14: "Elliptical contour lines indicate significant interactions between two factors while round contour lines correspond to low significance" Add the following reference:

  1. Evaluation of physicochemical characteristics and health risk of polycyclic aromatic hydrocarbons in borehole waters around automobile workshops in Southeastern Nigeria. Groundwater for Sustainable Development 14, 100615.

Reviewer 2 Report

In this manuscript, the authors extracted flavonoids from AS using deep eutectic solvents. Overall, the significance of this emerging field of green solvents for the extraction of bioactive compounds is of high interest to the scientific community.

The authors extensively looked into not only the extraction efficiency but also demonstrated how the physicochemical properties of these solvents affected their efficiency.

Minor comments for the authors for consistency:

  1. throughout the manuscript, hydrogen bond donors and hydrogen bond acceptors should be addressed as such but not only as hydrogen acceptors or hydrogen donors. (eg. Line 101 and 102)
  2. all scientific names should be written in italicized form when necessary (see lines 62 and 68)
  3. line 51, reference the first authors that introduced the NADES concept. Also, line 144 introduced this DES as hydrophobic DES. Please reference or demonstrate the physicochemical properties that support this DES to be hydrophobic DES (HDES).
  4. Line 72, discovered could be replaced with explored.
  5. Levulinic acid and Lactic acid could be confused with LA abbreviation, please address if necessary
  6. what accounts for the low extractability of this DES-8, although the viscosity was not very high? Could you explain this as part of your discussion?
  7. line 352- article "the" not "he"
  8. Figure 6 data in table 4, no need for the figure again or represent only in figure and exclude that data from the table
